# Ciliary neuropeptidergic signaling dynamically regulates excitatory synapses in postnatal neocortical pyramidal neurons

Lauren Tereshko, Ya Gao, Brian A Cary, Gina G Turrigiano*, Piali Sengupta*

Department of Biology, Brandeis University, Waltham, United States

**Abstract** Primary cilia are compartmentalized sensory organelles present on the majority of neurons in the mammalian brain throughout adulthood. Recent evidence suggests that cilia regulate multiple aspects of neuronal development, including the maintenance of neuronal connectivity. However, whether ciliary signals can dynamically modulate postnatal circuit excitability is unknown. Here we show that acute cell-autonomous knockdown of ciliary signaling rapidly strengthens glutamatergic inputs onto cultured rat neocortical pyramidal neurons and increases spontaneous firing. This increased excitability occurs without changes to passive neuronal properties or intrinsic excitability. Further, the neuropeptide receptor somatostatin receptor 3 (SSTR3) is localized nearly exclusively to excitatory neuron cilia both *in vivo* and in culture, and pharmacological manipulation of SSTR3 signaling bidirectionally modulates excitatory synaptic inputs onto these neurons. Our results indicate that ciliary neuropeptidergic signaling dynamically modulates excitatory synapses and suggest that defects in this regulation may underlie a subset of behavioral and cognitive disorders associated with ciliopathies.

**\*For correspondence:**
turrigiano@brandeis.edu (GGT);
sengupta@brandeis.edu (PS)

**Competing interest:** See
page 19

**Reviewing editor:** Anne E West,
Duke University School of
Medicine, United States

## Introduction

Primary cilia are microtubule-based compartmentalized organelles that are present on nearly all mammalian cell types including neurons (*Gerdes et al., 2009*; *Louvi and Grove, 2011*). Cilia concentrate signaling molecules and play critical roles in transducing environmental stimuli to regulate cellular properties (*Bangs and Anderson, 2017*; *Elliott and Brugmann, 2019*; *Goetz and Anderson, 2010*; *Hilgendorf et al., 2016*). Consequently, disruption of cilia and cilia-based signaling is causal to a set of pleiotropic disorders termed ciliopathies (*Davis and Katsanis, 2012*; *Reiter and Leroux, 2017*; *Youn and Han, 2018*). Abnormalities in brain development are a characteristic feature of many ciliopathies, highlighting the critical role of cilia in the nervous system (*Guemez-Gamboa et al., 2014*; *Louvi and Grove, 2011*; *Valente et al., 2014*; *Youn and Han, 2018*). Cilia have now been implicated in neurogenesis, neuronal migration, and establishment of synaptic connectivity during development (*Baudoin et al., 2012*; *Chizhikov et al., 2007*; *Guo et al., 2017*; *Guo et al., 2019*; *Higginbotham et al., 2012*; *Higginbotham et al., 2013*; *Lee et al., 2020*; *Spassky et al., 2008*; *Willaredt et al., 2008*). Intriguingly, cilia along with their complex signaling machinery are retained on mature neurons (*Arellano et al., 2012*; *Guadiana et al., 2016*; *Sterpka and Chen, 2018*), but whether ciliary signaling dynamically modulates mature neuronal properties has not been explored.

Recent studies have begun to implicate cilia in the establishment as well as maintenance of circuit connectivity and excitability in the postnatal brain. Loss of cilia and ciliary signaling results in defects in dendritic development and integration of adult born neurons into hippocampal circuits (*Kumamoto et al., 2012*). Disruption of ciliary signaling also reduces dendritic complexity and

affects synaptic connectivity of interneurons in the postnatal striatum (*Guo et al., 2017*). Moreover, cilia loss in mature dentate granule cells leads to altered contextual memory and synaptic plasticity at hippocampal mossy fiber synapses (*Rhee et al., 2016*). In a particularly interesting study, cilia in cerebellar Purkinje neurons were shown to be necessary for the maintenance of excitatory contacts from the climbing fibers of neurons in the inferior olivary nuclei of the medulla (*Bowie and Goetz, 2020*). In these reports, the effects on neuronal and circuit properties, and on synaptogenesis and synapse maintenance, manifested after prolonged (weeks to months) loss of cilia and/or ciliary signaling in the postnatal brain. Whether ciliary signaling can modulate synaptic properties on a more rapid timescale to adjust neuron and circuit excitability is unknown.

Neuronal cilia in different brain regions specifically localize a diverse set of neuropeptide and neurotransmitter receptors. Cilia-localized receptors in the brain include serotonin receptor 6, melanin-concentrating hormone receptor 1, somatostatin receptor 3 (SSTR3), and dopamine receptors D1, D2, and D5 among others (*Berbari et al., 2008a*; *Brailov et al., 2000*; *Domire et al., 2011*; *Hamon, 1999*; *Händel et al., 1999*; *Loktev and Jackson, 2013*; *Marley and von Zastrow, 2010*; *Schulz et al., 2000*). Signaling via these ciliary receptors is proposed to be mediated in part via regulation of adenylyl cyclase 3 (AC3) and the cAMP second messenger. Similar to the localization patterns of these receptors, AC3 is also specifically enriched in the cilia of diverse neuron types in the brain (*Berbari et al., 2007*; *Guadiana et al., 2016*; *Bishop et al., 2007*). Mutations in these receptors and AC3 are associated with a range of cognitive, metabolic, and behavioral disorders that are hallmarks of many ciliopathies (*Chen et al., 2016*; *Einstein et al., 2010*; *Lee and Gleeson, 2011*; *Loktev and Jackson, 2013*; *Wang et al., 2011*; *Youn and Han, 2018*). These receptors continue to be expressed in neuronal cilia in adults, and many of the cognate neurotransmitter and neuropeptide ligands are released locally by neurons or modulatory inputs, suggesting that cell–to-cell signaling through cilia-localized receptors plays an important role in the postnatal brain.

Neuronal circuits must maintain excitability within narrow bounds to prevent signal saturation or silencing (*Turrigiano and Nelson, 2000*), but the mechanisms that establish and dynamically maintain circuit excitability over a wide range of temporal and spatial scales are incompletely understood (*Turrigiano, 2017*). Several ciliopathies manifest with symptoms consistent with imbalances in excitability, such as cognitive impairment and recurrent epileptic seizures (*Guemez-Gamboa et al., 2014*; *Lee and Gleeson, 2011*; *Novarino et al., 2011*). These observations suggest the intriguing hypothesis that ciliary signaling plays an important role in adjusting neuronal excitability, either by altering intrinsic excitability through modulation of ion channel function or distribution, or by regulating the properties of excitatory or inhibitory synapses. The notion that neuropeptide and neurotransmitter release might converge on cilia to dynamically adjust intrinsic or synaptic properties, and thus modulate circuit excitability, has not been tested.

Here, we show that disruption of cilia and ciliary signaling in individual postnatal cortical pyramidal neurons in primary neuronal culture rapidly (<24 hr) and cell-autonomously strengthens excitatory synapses onto these neurons. Consistent with enhanced excitatory transmission, acute loss of cilia results in increased neuronal firing without affecting intrinsic neuronal excitability. In contrast to previous findings from chronic cilia disruption, acute cilia disruption has no major impact on dendritic arborization. We find that the SSTR3 neuropeptide receptor is specifically localized to the cilia of excitatory cortical neurons but not inhibitory interneuron subtypes, and that an SSTR3-selective antagonist and agonist bidirectionally modulate excitatory synaptic properties over similarly rapid timescales. Our results indicate that neuropeptidergic signaling via cilia-localized receptors dynamically modulates synaptic strength and plays a critical role in regulating neuronal excitability in the postnatal mammalian brain.

## Results

### Neuronal morphology is unaffected upon acute disruption of ciliary signaling in the postnatal cortex

Ciliogenesis in neocortical pyramidal neurons occurs progressively during early postnatal development, beginning at birth, with cilia reaching maximal lengths after several weeks (*Arellano et al., 2012*). To assess the development of cilia in cultured postnatal cortical neurons, we dissociated neurons from visual cortex of Long-Evans rat pups at postnatal days 0–1 (P0-1), and plated them onto

beds of confluent astrocytes as described previously (*Pratt et al., 2003*; *Tatavarty et al., 2020*). After 11 days in vitro (DIV), the majority of cortical neurons extended a single primary cilium from their soma, as assessed via staining with the neuronal cilia markers ARL13B and AC3 (*Figure 1A, B*; *Berbari et al., 2007*; *Bishop et al., 2007*; *Caspary et al., 2007*; *Sipos et al., 2018*). To ask whether cilia distribution and length are similar between excitatory and inhibitory neurons, we quantified the percentage of GAD67-positive (inhibitory interneurons) and GAD67-negative (excitatory) neurons containing cilia. Approximately 90% of both cell types exhibited cilia of lengths similar to those of postnatal neocortical neurons in vivo (*Arellano et al., 2012*; *Figure 1A–C*). Cilia lengths varied as expected (*Arellano et al., 2012*), but were similar across cell types (*Figure 1C*). These observations indicate that both excitatory and inhibitory postnatal cortical neurons contain primary cilia in culture at DIV11.

Since dendritic arbors are dynamic in these young postsynaptic neurons (*Pratt et al., 2008*; *Pratt et al., 2003*), we asked whether acute perturbation of ciliary signaling impacts dendritic morphology. To perturb cilia in a cell-autonomous manner, we transfected DIV9-10 cortical cultures at low efficiency (5–10 neurons transfected/dish) with GFP alone, or GFP and one of two shRNAs targeting the ciliary small GTPase Arl13b (shArl13b_1 and shArl13b_2). In this and all further experiments, GFP was used to identify and target transfected neurons for immunohistochemical or electrophysiological analysis. Mutations in Arl13b have been shown to affect ciliary signaling without fully truncating cilia (*Caspary et al., 2007*; *Cevik et al., 2010*; *Larkins et al., 2011*; *Lu et al., 2015*). Pyramidal neurons transfected with either shRNA for 24 or 48 hr showed a reduction in total immunolabeled ARL13B fluorescence in cilia by ~50% as compared to non-transfected control cells (*Figure 1D,E*, *Figure 1—figure supplement 1A*), indicating that both shRNAs were effective at rapidly knocking down ARL13B (henceforth referred to as acute knockdown). This reduction in ARL13B was sufficient to shorten cilia (assessed using AC3 fluorescence) in these postnatal pyramidal neurons (*Figure 1F*, *Figure 1—figure supplement 1B*), as reported previously in other cell types (*Caspary et al., 2007*; *Cevik et al., 2010*; *Larkins et al., 2011*; *Lu et al., 2015*). However, in contrast to the significant reduction in dendritic complexity observed upon prolonged conditional Arl13b deletion in striatal interneurons (*Guo et al., 2017*), acute and cell-autonomous knockdown of ARL13B for 24 or 48 hr had no impact on the total length of apical-like dendrites, or on the number of dendritic branch points (*Figure 1G–I*, *Figure 1—figure supplement 1C–E*).

Since knockdown of ARL13B affects ciliary signaling but does not fully truncate cilia (*Caspary et al., 2007*; *Higginbotham et al., 2012*; *Higginbotham et al., 2013*), we tested whether more severe disruption of cilia structure is sufficient to rapidly alter neuronal morphology. The basal body component CEP164 and the intraflagellar transport protein IFT88 are essential for ciliogenesis and cilia maintenance (*Graser et al., 2007*; *Pazour et al., 2000*). shRNA-mediated knockdown of either IFT88 or CEP164 alone led to only a modest knockdown even after 48 hr with a concomitant weak effect on cilia length (*Figure 1—figure supplement 1F,G*). However, while co-expression of shIft88 and shCep164 reduced IFT88 immunofluorescence by ~30% after 24 hr, co-transfection of both shRNAs reduced IFT88 immunofluorescence by ~70% after 48 hr (*Figure 1—figure supplement 1F,H*). Consistently, co-transfection of shIft88 and shCep164 also resulted in severely truncated cilia after 48 hr of transfection (*Figure 1—figure supplement 1G,H*). Despite the dramatic disruption of cilia structure under these conditions, we again observed no gross effects on pyramidal neuron morphology (*Figure 1—figure supplement 1C–E*). We conclude that acute cilia disruption does not strongly impact short-term maintenance of postnatal pyramidal neuron dendritic morphology.

## Acute knockdown of ARL13B selectively strengthens excitatory synapses

Conditional, prolonged depletion of ARL13B at postnatal stages alters not only morphology but also connectivity of striatal interneurons (*Guo et al., 2017*). Moreover, cilia disruption for weeks to months induces loss of climbing fibers synapses onto Purkinje cells and reduces synaptic integration of adult-born dentate granule cells (*Bowie and Goetz, 2020*; *Kumamoto et al., 2012*). We wondered whether cell-autonomous disruption of cilia function is sufficient to regulate the strength or number of excitatory and/or inhibitory synapses on a more rapid timescale (24–48 hr).

To address this issue, we acutely disrupted cilia via transfection of shArl13b, or co-transfection of shCep164 and shIft88, into DIV9-10 cultured pyramidal neurons. We then fixed and immunostained

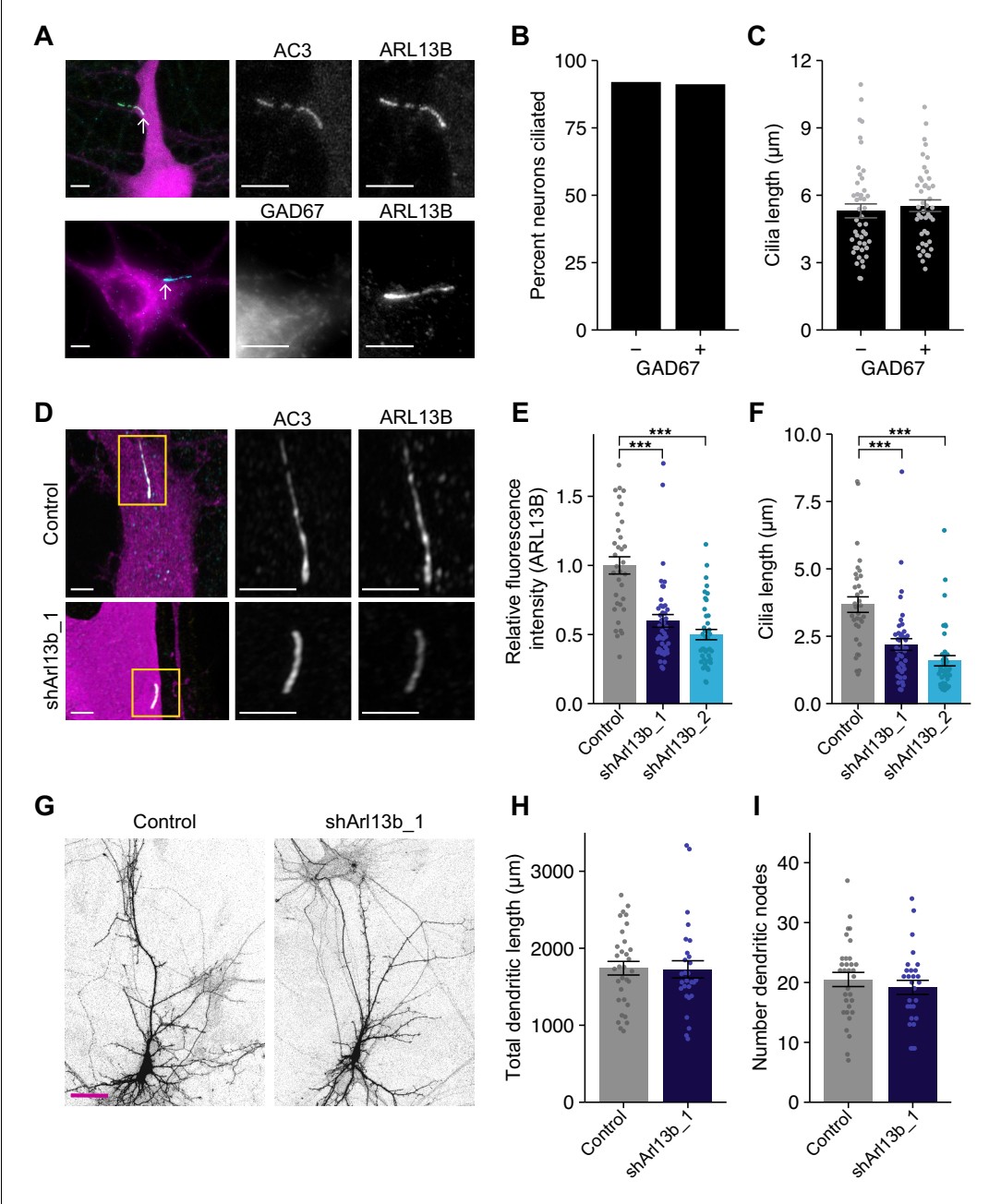

**Figure 1.** Acute knockdown of ciliary proteins does not alter dendritic morphology of cortical pyramidal neurons. (**A**) Representative images of a DIV11 pyramidal neuron expressing GFP (top), and an inhibitory neuron immunolabeled with GAD67 antibodies (bottom). Cilia (arrows) are immunolabeled using antibodies against AC3 and ARL13B (top) or ARL13B alone (bottom). Scale bars: 5 µm. (**B**) Percentage of inhibitory (GAD67+) and excitatory (GAD67−) neurons containing cilia immunolabeled with antibodies against AC3 and/or ARL13B. n = 150 total; three dissociations. (**C**) Lengths of cilia in excitatory and inhibitory neurons. Each dot is a measurement from a single neuron. Bars are average ± SEM. n: GAD67+ = 44, GAD67− = 46; five dissociations. (**D**) Representative images of neurons expressing GFP alone (top) or shArl13b_1 and GFP (see Key Resources) (bottom). Cilia were immunolabeled with antibodies against AC3 and ARL13B. Images at right show enlarged (2.5×) views of cilia (yellow boxes). Scale bars: 5 µm. (**E and F**) Relative fluorescence intensities of immunolabeled ciliary ARL13B (**E**) and cilia lengths (**F**) in neurons transfected with the indicated plasmids. Each dot is a measurement from a single neuron. Values in (**E**) are normalized to intensity in control neurons. Bars are average ± SEM. *** indicates p<0.001 for the indicated conditions (Kruskal–Wallis with Dunn's multiple comparisons test). n: Control = 34, shArl13b_1 = 43, shArl13b_2 = 40; four dissociations. (**G**) Representative images of pyramidal neurons expressing GFP alone (control) or shArl13b_1 and GFP. Scale bar: 50 µm. (**H and I**) Total lengths (**H**) and number of branch points (**I**) of apical-like dendritic arbors of neurons expressing GFP or shArl13b_1 and GFP. Each dot is a measurement from a single neuron. Bars are average ± SEM. n: Control = 32, shArl13b_1 = 28; four dissociations. Here and below, statistical tests used and exact p-values for each comparison are shown in *Supplementary file 1*. Also see *Figure 1—figure supplement 1*.

*Figure 1 continued on next page*

*Figure 1 continued*

The online version of this article includes the following source data and figure supplement(s) for figure 1:

**Source data 1.** Source data for *Figure 1*.
**Figure supplement 1.** Gross neuronal morphology is unaltered upon acute knockdown of ciliary proteins.
**Figure supplement 1—source data 1.** Source data for *Figure 1—figure supplement 1*.

these cultures after 24 or 48 hr using antibodies against the excitatory presynaptic marker VGlut1, and the postsynaptic AMPA-type glutamate receptor (AMPAR) subunit GluA2 under non-permeant conditions to label surface receptors; sites of colocalization are considered putative excitatory synaptic sites (*Figure 2A,B*). Manipulation of cilia signaling using either method increased the intensity of the surface synaptic GluA2 signal; this increase was evident at both 24 hr and 48 hr after transfection with shArl13b, and at 48 hr after transfection with shIft88 and shCep164 (*Figure 2C*), indicating that reducing ciliary function increases the synaptic accumulation of AMPAR. Postsynaptic reduction of cilia function with shArl13b_1 also increased presynaptic expression of VGlut1, although this change was less robust (*Figure 2—figure supplement 1A*). Quantification of the density of putative excitatory synapses along dendritic arbors also revealed a significant increase in excitatory synapse density that was evident at 48 hr after knockdown (*Figure 2D*). Thus, acute and cell-autonomous cilia disruption increases both the number of excitatory synapses and the accumulation of synaptic AMPAR. Together, these changes are predicted to enhance excitatory synaptic drive.

Since the balance between excitation and inhibition (E/I balance) is determined by the relative drive from glutamatergic and GABAergic neurons, we next asked whether ciliary signaling also impacts inhibitory synapses. Different inhibitory interneuron subtypes preferentially synapse onto different compartments of cortical pyramidal neurons (*Kepecs and Fishell, 2014*; *Tremblay et al., 2016*; *Urban-Ciecko and Barth, 2016*). Somatic synapses are difficult to quantify in culture due to the density of somata, so we focused on the more readily quantifiable inhibitory synapses that contact the apical-like dendrites of pyramidal neurons. We used colocalization of GAD65 (a presynaptic inhibitory marker) and Gephyrin (a postsynaptic inhibitory marker) to identify putative inhibitory synapses (*Figure 2E*); we observed no significant change in the fluorescence intensities of either marker after either 24 or 48 hr following transfection with either shArl13b alone or shIft88 and shCep164 together (*Figure 2E,F*, *Figure 2—figure supplement 1B*). Inhibitory synapse density was also unaltered following acute cilia disruption at either time point (*Figure 2G*). We infer that under our experimental conditions, ciliary signaling acutely regulates excitatory but not inhibitory synapses onto cultured postnatal pyramidal neurons.

## Acute knockdown of ARL13B increases AMPAR-mediated glutamatergic currents

Since fast glutamatergic transmission is mainly mediated by AMPAR (*Huganir and Nicoll, 2013*; *Traynelis et al., 2010*), increased accumulation of this receptor at the postsynaptic membrane is predicted to increase the strength of excitatory synapses. We tested this by recording AMPAR-mediated miniature excitatory postsynaptic currents (mEPSCs), which represent the postsynaptic response to release of individual vesicles of glutamate; the amplitude of these currents is a direct correlate of postsynaptic strength. To isolate and measure mEPSCs, we obtained whole cell voltage clamp recordings from DIV11 control or shArl13b-transfected neurons in the presence of tetrodotoxin (to block spike-mediated release), APV (to block NMDA receptor-mediated currents), and picrotoxin (to block GABA$_A$-mediated currents) (*Figure 3A*, left). Consistent with the increased accumulation of synaptic AMPAR, neurons whose cilia were acutely disrupted had larger AMPAR-mediated mEPSCs compared to transfected controls (*Figure 3A*, right). Analysis of the cumulative probability distribution function of individual events showed that both manipulations shifted the amplitude distributions toward larger values (*Figure 3B*). mEPSC frequency is quite variable in culture and was not significantly affected by cilia disruption at 24 hr (*Figure 3C*). The increase in mEPSC amplitude caused by cilia disruption was not caused by changes in passive electrical properties, as input resistance (which can affect voltage clamp efficacy) and resting potentials (a measure of cell health) were unaltered (*Figure 3D,E*). These results indicate that ciliary signaling acts cell-autonomously to rapidly increase excitatory postsynaptic strength.

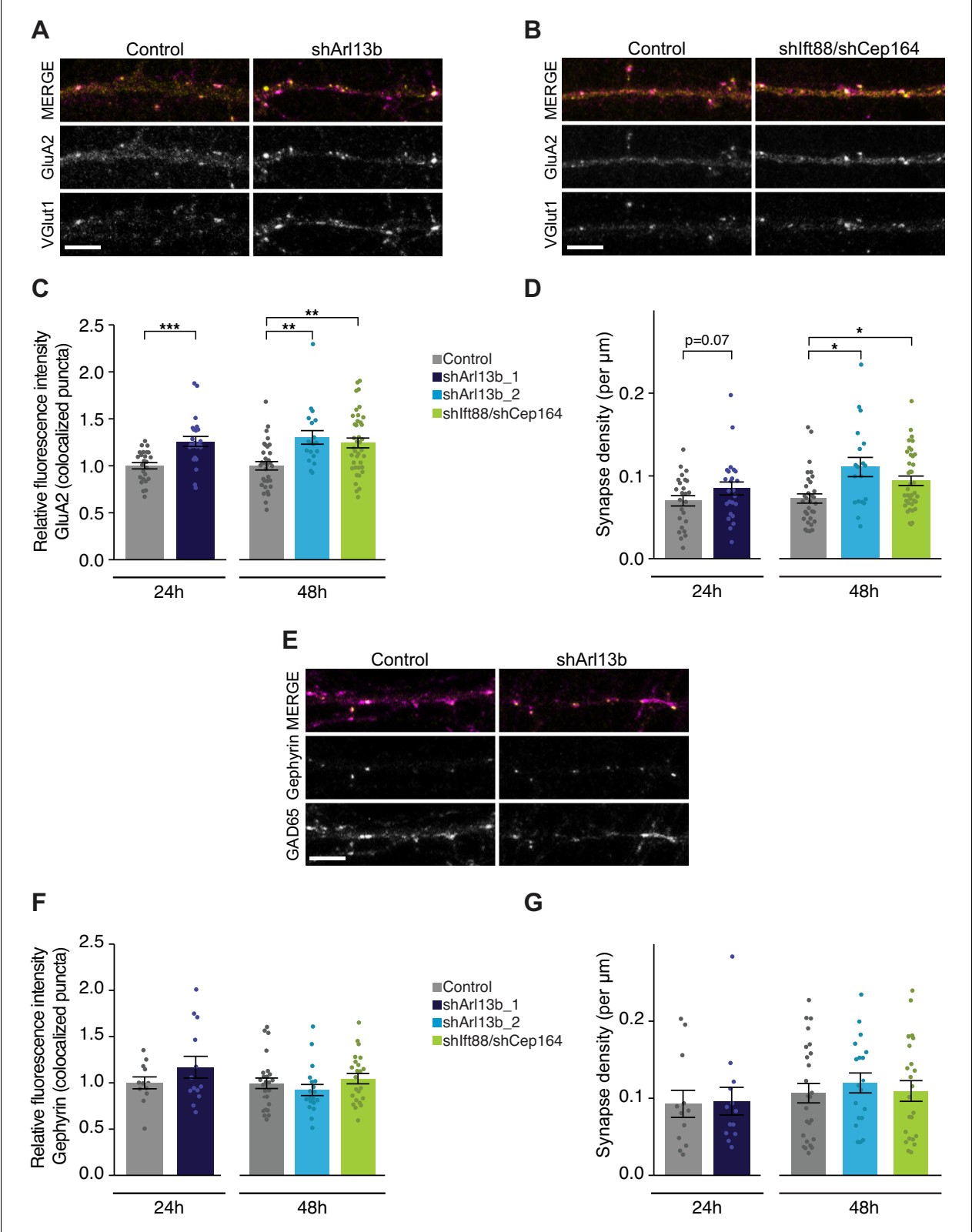

**Figure 2.** Acute knockdown of ciliary proteins increases the strength and number of excitatory synapses. (A, B, and E) Representative images of pyramidal neuron dendrites immunolabeled with antibodies against GluA2 and VGlut1 (**A and B**), or Gephyrin and GAD65 (**E**). Cultures were transfected with GFP alone or together with the indicated plasmids. Scale bars: 5 µm. (C) Relative fluorescence intensity of immunolabeled GluA2 at GluA2 /VGlut1 colocalized puncta for indicated conditions at 24 hr or 48 hr following transfection. Intensity values are normalized to controls. Each dot

*Figure 2 continued on next page*

*Figure 2 continued*

is the average summed pixel value for all measured synapses from a given neuron. Bars are average ± SEM. ** and *** indicate p<0.01 and 0.001, respectively, for the indicated conditions (LMM with Dunnett-type correction for multiple comparisons). n: Control = 25 (24 hr) and 32 (48 hr), shArl13b_1 = 24, shArl13b_2 = 19, shIft88/shCep164 = 39; four dissociations. (D) Number of colocalized GluA2/VGlut1 puncta per µm of dendrite analyzed (density) onto neurons transfected with the indicated plasmids at 24 hr or 48 hr following transfection. Each dot is the density of synapses examined per neuron. Bars are average ± SEM. * indicates p<0.05 for the indicated conditions (LMM with Dunnett-type correction for multiple comparisons); additional p-values are also indicated. n: As in C. (F) Relative fluorescence intensity of immunolabeled Gephyrin at colocalized puncta on neurons transfected with the indicated plasmids at 24 hr or 48 hr following transfection. Intensity values are normalized to values in control neurons. Each dot is the average summed pixel value for all measured synapses from a given neuron. Bars are average ± SEM. n: Control = 17 (24 hr), shArl13b_1 = 22 neurons; four dissociations; and Control = 25 (48 hr), shArl13b_2 = 19, shIft88/shCep164 = 23; three dissociations. (G) Number of colocalized Gephyrin/GAD65 puncta per µm of dendrite analyzed (density) onto neurons transfected with the indicated plasmids at 24 hr or 48 hr following transfection. Each dot is the density of synapses examined per neuron. Bars are average ± SEM. n: As in F. Also see *Figure 2—figure supplement 1*.

The online version of this article includes the following source data and figure supplement(s) for figure 2:

**Source data 1.** Source data for *Figure 2*.

**Figure supplement 1.** Presynaptic VGlut1 staining is increased upon acute knockdown of ciliary proteins.

**Figure supplement 1—source data 1.** Source data for *Figure 2—figure supplement 1*.

## Acute knockdown of ARL13B increases spontaneous firing without affecting intrinsic excitability

Increasing excitatory synapse number and strength without a concomitant change in inhibitory synapses would be expected to increase net excitatory synaptic drive and elicit more action potentials. To investigate whether knockdown of ARL13B increases firing, we performed whole cell patch clamp recordings in current clamp under conditions where network activity was intact and synaptic drive can elicit action potentials (*Figure 4A*). To compensate for differences in resting potential across neurons, a small DC current was injected to maintain the inter-spike membrane potential close to −55 mV (see Materials and methods). We targeted control or knockdown GFP-transfected neurons, recorded firing driven by network activity, and calculated the mean firing rate. As expected, we observed a broad distribution of mean firing rates (*Trojanowski et al., 2021*; *Turrigiano et al., 1998*); this distribution was shifted toward larger values after acute ARL13B knockdown, such that mean firing rate roughly doubled upon this manipulation (*Figure 4A*, right).

Spontaneous firing could also be enhanced if the intrinsic excitability of neurons was increased by cilia disruption. Intrinsic excitability is controlled by the balance of voltage-gated ion channels in the cell membrane which determine how many action potentials a neuron fires for a given amount of depolarizing current. To determine whether cilia disruption impacts the intrinsic excitability of pyramidal neurons, we pharmacologically blocked excitatory and inhibitory synaptic currents (see Materials and methods), injected direct current steps to evoke spikes, and then plotted the number of spikes evoked as a function of injected current to generate firing rate vs current (F–I) curves. Knockdown of ARL13B had no significant impact on intrinsic excitability (*Figure 4B*). Taken together with the lack of an effect on passive neuronal properties and dendritic arborization, these data show that intrinsic neuronal excitability and morphology are unaffected by acute and cell-autonomous manipulation of cilia. Instead, the increase in mean firing rate is likely driven by the increase in number and strength of excitatory synapses.

## The SSTR3 neuropeptide receptor is largely restricted to the cilia of excitatory neurons in the postnatal cortex

Cilia specifically localize multiple neuropeptide receptors, a subset of which has been implicated in the regulation of neuronal and circuit properties in the developing and mature brain (*Einstein et al., 2010*; *Guo et al., 2017*; *Hilgendorf et al., 2016*; *Loktev and Jackson, 2013*; *Mykytyn and Askwith, 2017*). In particular, the somatostatin receptor 3 (SSTR3) is widely expressed in the brain and is commonly employed as a marker for neuronal cilia (*Berbari et al., 2007*; *Guadiana et al., 2016*; *Händel et al., 1999*; *Schulz et al., 2000*; *Stanić et al., 2009*). Moreover, its endogenous ligand somatostatin is expressed by a subset of cortical interneurons (*Gonchar et al., 2007*; *Lee et al., 2010*; *Xu et al., 2010*). However, the role of somatostatin and SSTR3-mediated signaling in the neocortex is largely uncharacterized.

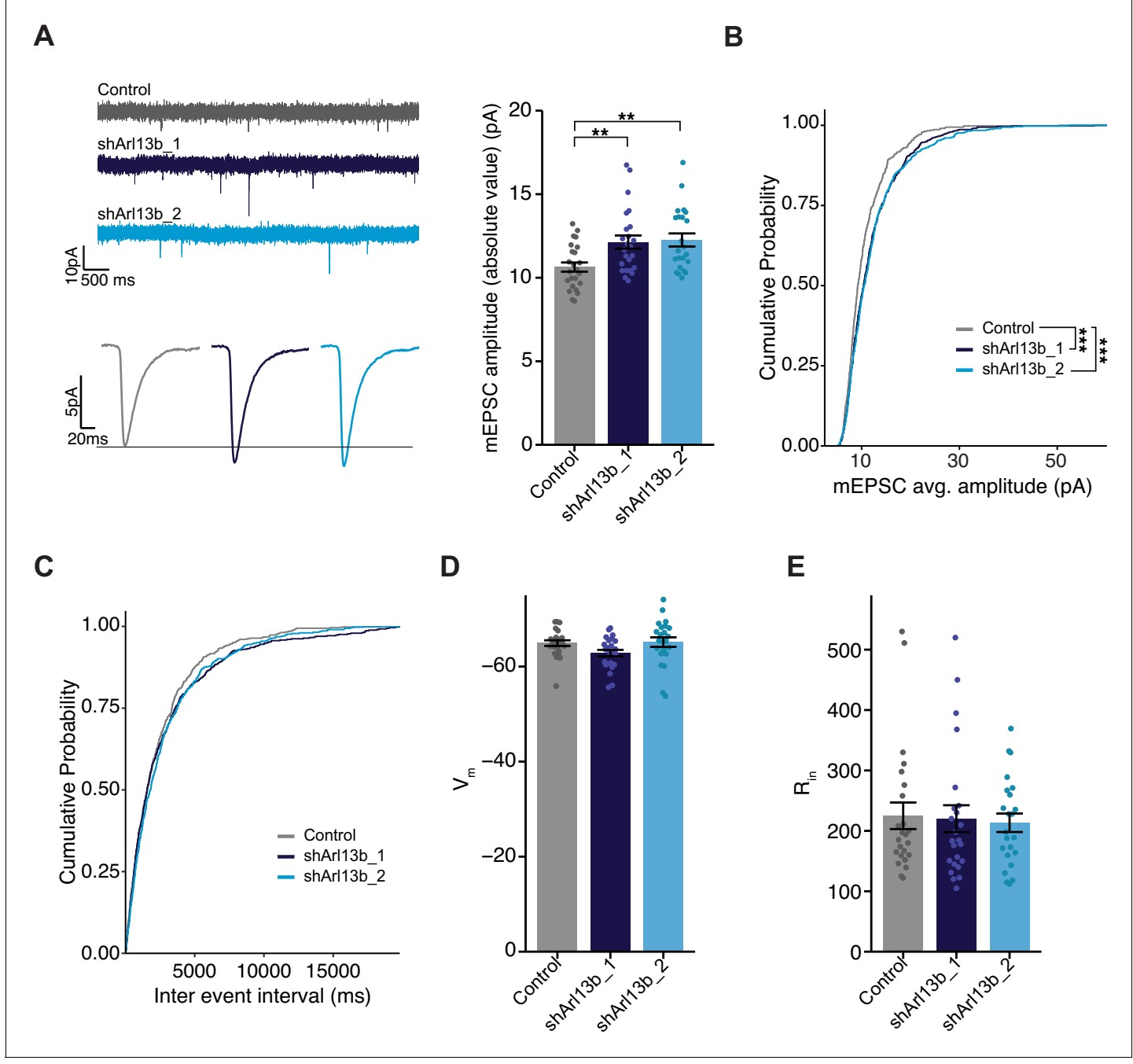

**Figure 3.** Mean mEPSC amplitude in pyramidal neurons is increased following acute reduction of ciliary signaling. (**A**) (Left) Representative mEPSC traces (top) and average waveforms (bottom) from neurons transfected with the indicated plasmids. (Right) mEPSC amplitude from neurons transfected with the indicated plasmids. Each dot represents the average amplitude for a given neuron. Bars are average ± SEM. ** indicates the difference between indicated values at p<0.01 (Kruskal–Wallis with Dunn's multiple comparisons test). n: Control = 24, shArl13b_1 = 24, shArl13b_2 = 23; >5 dissociations. (**B** and **C**) Cumulative distribution probabilities of mEPSC amplitudes (**B**) and inter-event intervals (**C**) from neurons transfected with the indicated plasmids. *** indicates the difference from control at p<0.001 (Kruskal–Wallis with Bonferroni correction). (**D** and **E**) Average resting membrane potential ($V_m$) (**D**) and input resistance ($R_{in}$) (**E**) of neurons transfected with the indicated plasmids. Each dot represents a single neuron. Bars are average ± SEM. n: As in **A**.

The online version of this article includes the following source data for figure 3:

**Source data 1.** Source data for *Figure 3*.

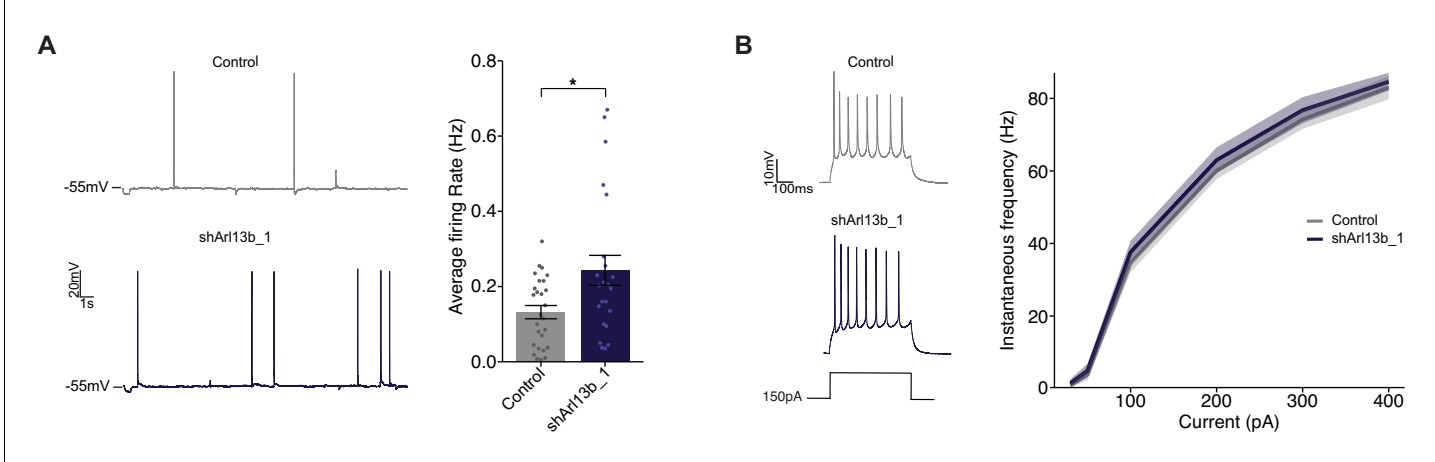

**Figure 4.** Disruption of ciliary signaling increases spontaneous firing without influencing intrinsic excitability. (A) (Left) Representative voltage traces of spontaneous activity recorded from pyramidal neurons expressing GFP alone or shArl13b_1 and GFP. (Right) Average spontaneous firing rate for neurons transfected with the indicated plasmids. Each dot represents one neuron. Bars are average ± SEM. * indicates different between indicated values at p<0.05 (Wilcoxon rank-sum test). n: Control = 32, shArl13b_1 = 25; >5 dissociations. (B) (Left) Representative responses of pyramidal neurons expressing GFP or shArl13b_1 and GFP during current injection. (Right) Average instantaneous firing rate vs current curves for neuron transfected with the indicated plasmids. Errors are SEM. n: Control = 18, shArl13b_1 = 20; three dissociations.

The online version of this article includes the following source data for figure 4:

**Source data 1.** Source data for *Figure 4*.

Expression of SSTR3 begins at birth and increases during postnatal development in the rat hippocampus (*Stanić et al., 2009*). To begin examining a possible role for ciliary SSTR3 mediated signaling in regulating excitatory synapses in the cortex, we first characterized the expression and localization of this receptor in primary visual cortex in vivo. P15–16 animals were injected with a GFP-expressing AAV viral vector, and 7 days later, brain slices were examined via immunostaining. GFP-expressing pyramidal neurons were identified by their characteristic morphologies, and primary cilia were co-labeled with antibodies against AC3 and SSTR3. We found that the majority of pyramidal neurons in each cortical layer contained cilia positive for both AC3 and SSTR3, with a small subset of cilia expressing AC3 alone (*Figure 5A,B*). To further assess the expression of SSTR3 in neuronal populations, we immunostained fixed cortical slices for the neuronal marker NeuN and inhibitory neuron marker GAD67 together with SSTR3. We found that SSTR3 was present in the cilia of many although not all neurons across layers (*Figure 5—figure supplement 1A,B*). In the neocortex, NeuN preferentially marks excitatory neurons (*Chattopadhyaya et al., 2004*). We noted that while the majority of NeuN-positive neurons expressed ciliary SSTR3 (*Figure 5—figure supplement 1A,B*), fewer than 10% of neurons that stained weakly with NeuN but were GAD67-positive in each layer expressed SSTR3 (*Figure 5—figure supplement 1A,B*). The cilia of GAD67-positive interneurons retained expression of AC3 (*Figure 5C*). These results indicate that ciliary SSTR3 localization is restricted largely to cortical excitatory neurons and is present in only a small fraction of inhibitory neurons.

We asked whether the small population of inhibitory neurons expressing ciliary SSTR3 represents a defined inhibitory interneuron subtype. To address this, we immunostained fixed cortical slices with antibodies against SSTR3 and the interneuron subtype-specific markers choline acetyltransferase (ChAT), parvalbumin (PV), somatostatin (SOM), and vasoactive intestinal peptide (VIP). We observed only rare (<5%) ciliary SSTR3 expression in inhibitory interneurons positive for PV and SOM (*Figure 5D*, *Figure 5—figure supplement 1C*), suggesting that the small number of SSTR3-positive GABAergic neurons is likely to be an alternate inhibitory neuron subtype(s).

To determine if dissociated cultures also expressed ciliary SSTR3, we co-immunostained cortical cultures for ARL13b and SSTR3. We noted that ~30% of ARL13B-positive cilia on cultured neurons were also positive for SSTR3 at DIV11 (*Figure 5E,F*), indicating that SSTR3 is expressed at detectable levels in a subset of these young neocortical neurons. The majority of SSTR3-expressing neurons in culture were excitatory, based on the absence of co-staining with the inhibitory neuron-

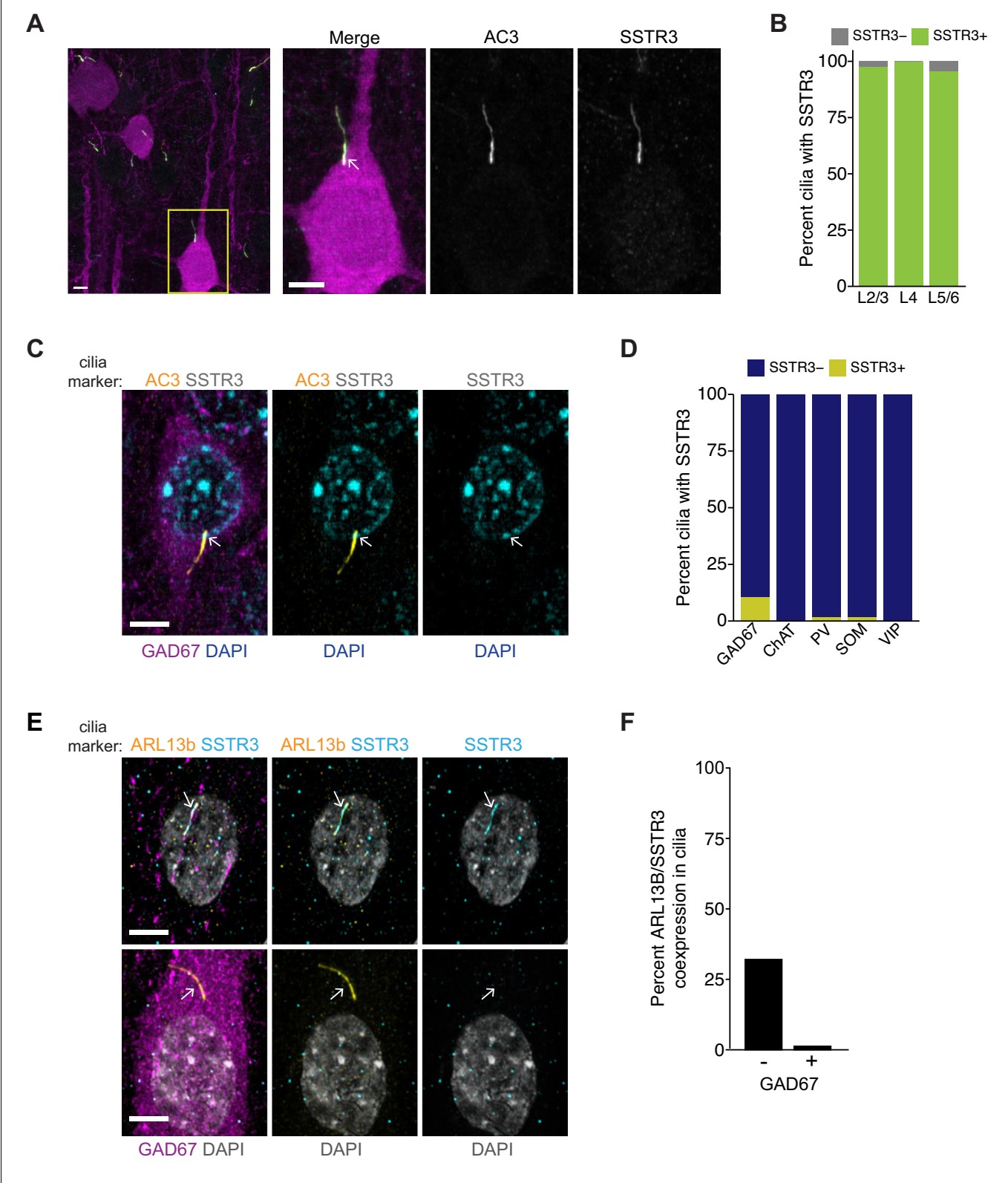

**Figure 5.** SSTR3 is localized to the cilia of excitatory, but not inhibitory, cortical neurons. (**A**) Representative images of primary cilia immunolabeled with antibodies against AC3 and SSTR3 in GFP-expressing neurons in fixed cortical slices from P22 animals. Images at right show enlarged (2.5×) views of cilium (yellow box; arrow). Scale bars: 5 μm. (**B**) Percentage of GFP-expressing pyramidal neurons with primary cilia co-expressing AC3 and SSTR3 categorized by cortical layer. n = 150 neurons per layer; three animals. (**C**) Representative images of a GAD67-expressing inhibitory neuron in fixed

*Figure 5 continued on next page*

*Figure 5 continued*

cortical tissue stained with DAPI. Cilia were immunolabeled with antibodies against AC3 and SSTR3. The cilium is indicated with an arrow. Scale bar: 5 μm. (**D**) Percentage of inhibitory neurons of the indicated subtype containing SSTR3+ primary cilia in fixed cortical tissue. Cilia were identified via co-immunostaining with anti-AC3 or anti-PCTN antibodies. n: GAD67+ = 115, ChAT+ = 31, PV+ = 100, SOM+ = 110; VIP+ = 150; three animals. (**E**) Representative images of cultured neurons immunolabeled with antibodies against GAD67, ARL13B, and SSTR3, and co-stained with DAPI. Arrows indicate cilia of GAD− (top) and GAD+ (bottom) neurons. Scale bars: 5 μm. (**F**) Quantification of cultured neurons immunolabeled with antibodies against GAD67, ARL13B, and SSTR3. n = 515 total; four dissociations. Also see *Figure 5—figure supplement 1*.

The online version of this article includes the following source data and figure supplement(s) for figure 5:

**Source data 1.** Source data for *Figure 5*.

**Figure supplement 1.** SSTR3 is present in the majority of the cilia of NeuN-positive neurons and absent from multiple inhibitory neuron subtypes in the cortex.

**Figure supplement 1—source data 1.** Source data for *Figure 5—figure supplement 1*.

specific marker GAD67 (*Figure 5E,F*). Importantly, SSTR3 was specifically localized to cilia in all expressing cells (*Figure 5E*). Taken together, these results indicate that SSTR3 is expressed primarily, if not exclusively, by excitatory neurons in the neocortex, and is localized specifically to their cilia both in vitro and in vivo.

## Ciliary SSTR3 signaling bidirectionally modulates excitatory synapses in culture

Given that SSTR3 is enriched in the cilia of cortical excitatory neurons, and somatostatin is present in, and released by, a subset of inhibitory GABAergic interneurons (*Gonchar et al., 2007*; *Lee et al., 2010*; *Xu et al., 2010*), we examined whether signaling via SSTR3 mediates cilia-dependent modulation of excitatory synapse strength. To manipulate SSTR3 signaling, we took advantage of the previously described SSTR3-selective agonist (L-796,778) and antagonist (MK-4256), which can bidirectionally regulate SSTR3-mediated signaling (*He et al., 2012*; *Rohrer et al., 1998*; *Rohrer and Schaeffer, 2000*) (see Materials and methods). We treated DIV10-11 neuronal cultures for varying periods of time with either the agonist or antagonist, and then fixed and stained for excitatory pre- and post-synaptic markers. Since not all neurons in culture express detectable SSTR3 in their cilia, we additionally immunostained with antibodies against SSTR3 and confined our analysis to neurons with robust SSTR3 expression. Because of species overlap between antibodies, we used Shank3 as the postsynaptic marker for these experiments; Shank3 specifically localizes to excitatory synapses and the intensity of synaptic Shank3 is correlated with postsynaptic strength (*Monteiro and Feng, 2017*; *Tatavarty et al., 2020*; *Verpelli et al., 2011*).

Manipulating SSTR3 signaling induced bidirectional changes in the intensity of both Shank3 and VGlut1 at colocalized sites that developed over several hours (*Figure 6A,B*, *Figure 6—figure supplement 1A*). The SSTR3 antagonist significantly increased synaptic Shank3 (*Figure 6A,B*) and VGlut1 intensity (*Figure 6—figure supplement 1A*), while the SSTR3 agonist significantly reduced the fluorescence intensities of both markers (*Figure 6A,B*, *Figure 6—figure supplement 1A*). The SSTR3 agonist also significantly reduced the density of putative excitatory synapses, while the effects of the antagonist on synapse density did not reach statistical significance (*Figure 6C*). Experiments were performed using the lowest concentrations of each compound at which synaptic effects were observed (*Figure 6—figure supplement 1B,C*). Taking advantage of the temporal resolution of these pharmacological experiments, we determined how rapidly these manipulations were able to modulate synaptic properties. We found that the first detectable effects were evident after 18 hr and were more robust after 24 hr of treatment (*Figure 6B,C*). Neither the agonist nor the antagonist altered cilia lengths (*Figure 6—figure supplement 1D*) or affected cell viability (*Figure 6—figure supplement 1E*). These results indicate that SSTR3-mediated signaling can dynamically regulate excitatory synaptic properties and suggest that this neuropeptidergic signaling pathway negatively regulates excitatory synaptic strength.

Although we detected SSTR3 only in the cilia of excitatory neurons, this receptor may nevertheless be present in, and function, elsewhere in the cell. We tested whether the effects of pharmacological manipulation of SSTR3 on synaptic strength are mediated through ciliary signaling. We knocked down cilia function, applied the SSTR3 agonist, and quantified the intensity of the excitatory synaptic marker Shank3 (*Figure 6B*). Neurons were transfected with GFP alone or with shIft88

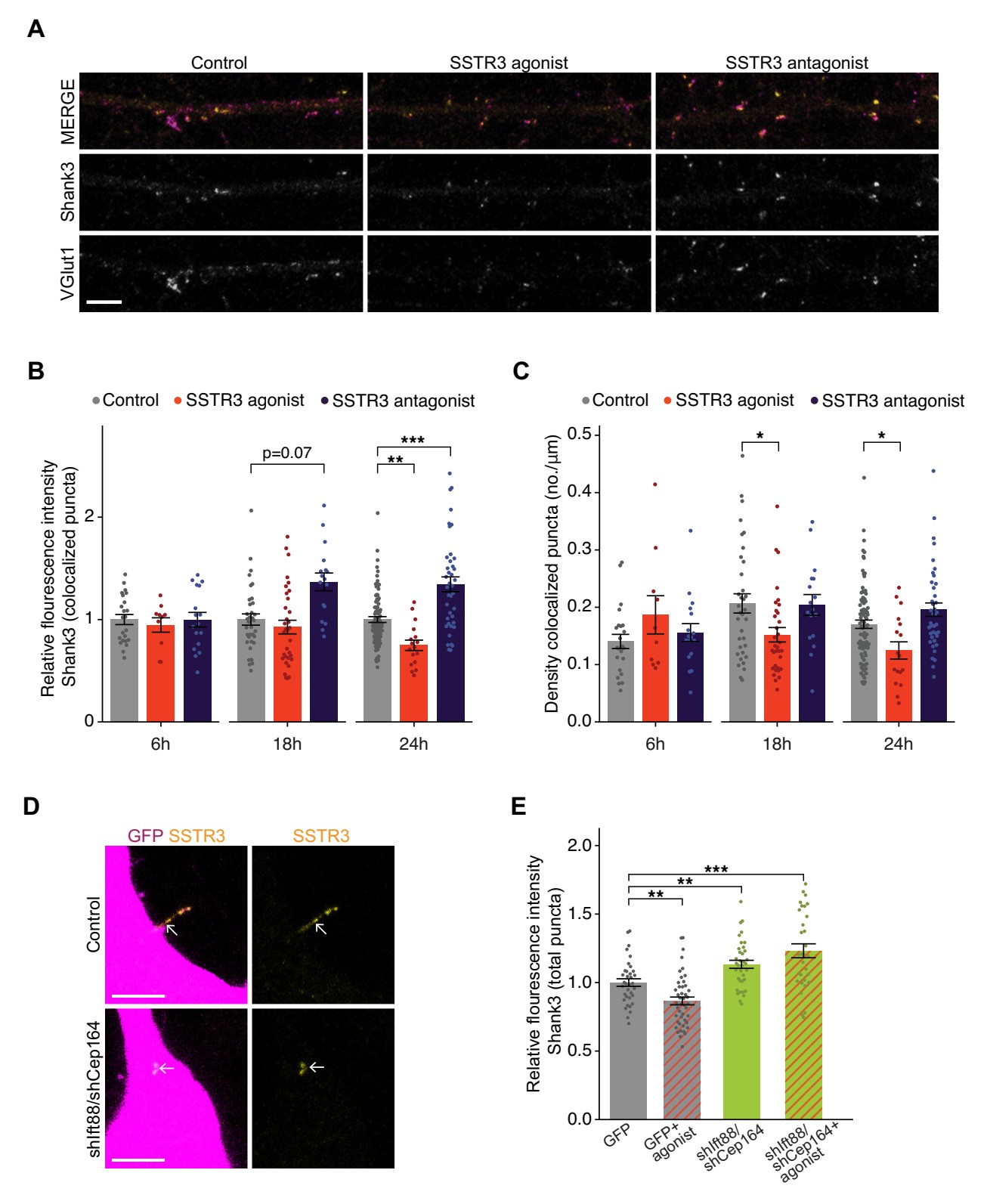

**Figure 6.** Pharmacological modulation of ciliary SSTR3 bidirectionally regulates excitatory synapses. (**A**) Representative images of excitatory neuron dendrites immunolabeled with antibodies against Shank3 (Shk3) and VGlut1. Cultures were treated for 24 hr with 2 μM L-796,798 (SSTR3 agonist; purified – see Materials and methods) or 1 μM MK-4256 (SSTR3 antagonist) prior to staining. Scale bars: 5 μm. (**B**) Relative fluorescence intensity of Shk3 at colocalized Shk3/VGlut1 puncta on neurons treated with the indicated compounds and fixed and immunostained 6, 18, or 24 hr after addition of

*Figure 6 continued on next page*

*Figure 6 continued*

drug. Intensity values are normalized to values in control neurons. Each dot is the average summed pixel value of all examined synapses per neuron. Bars are average ± SEM. ** and *** indicate p<0.01 and 0.001, respectively, for the indicated conditions (LMM with Dunnett-type correction for multiple comparisons); additional p-values are also shown. n: (6 hr) Control = 23, agonist = 10, antagonist = 17; (18 hr) Control = 35, agonist = 33, antagonist = 17; (24 hr) Control = 87, agonist = 17, antagonist = 40; ≥3 dissociations. (C) Number of colocalized Shk3/VGlut1 puncta per μm of dendrite analyzed (density) onto neurons treated with the indicated compounds. Cultures were immunostained at the indicated times following addition of the compounds. Each dot is the density of synapses examined per neuron. Bars are average ± SEM. * indicates p<0.05 for the indicated conditions (LMM with Dunnett-type correction for multiple comparisons). n: as in B. (D) Representative images of cilia (arrows) of control or shIft88/shCep164-transfected neurons immunolabeled with antibodies against SSTR3. Scale bars: 5 μm. (E) Relative fluorescence intensity of total Shk3 puncta on neurons transfected with the indicated constructs, then fixed and immunostained 24 hr after addition of DMSO or 2 μM L-796,798 (SSTR3 agonist). Intensity values are normalized to values in GFP+ control neurons. Each dot is the average summed pixel value of all examined puncta per neuron. Bars are average ± SEM. ** and *** indicate p<0.01 and 0.001, respectively, for the indicated conditions (Kruskal–Wallis with Dunn correction for multiple comparisons). n: GFP+ control = 35, GFP + agonist = 46, shIft88/shCep164/GFP = 36, shIft88/shCep164/GFP + agonist = 31; three dissociations. Also see *Figure 6—figure supplement 1*.

The online version of this article includes the following source data and figure supplement(s) for figure 6:

**Source data 1.** Source data for *Figure 6*.
**Figure supplement 1.** Agonism and antagonism of SSTR3 alter excitatory synaptic strengths.
**Figure supplement 1—source data 1.** Source data for *Figure 6—figure supplement 1*.

and shCep164 for 48 hr, treated with DMSO (control) or SSTR3 agonist for 24 hr, and were then fixed and immunolabeled with antibodies against Shank3 and SSTR3. SSTR3 labeling could be detected in many but not all of the shortened cilia of shRNA-transfected neurons (*Figure 6D*), indicating that this receptor retained localization to these truncated cilia. Consistent with our previous data set (*Figure 6B*), the SSTR3 agonist decreased the fluorescence intensity of Shank3 puncta in control (GFP-expressing) neurons (*Figure 6E*). However, in hairpin-expressing neurons with truncated cilia, the intensity of the Shank3 signal was increased (*Figure 6E*), also as expected (*Figure 2C*). Finally, the SSTR3 agonist was not able to reduce Shank3 intensity in neurons with truncated cilia (*Figure 6E*). Together, these data suggest that SSTR3-mediated modulation of excitatory synapses likely relies on proper cilia function.

## Discussion

We show here that cilia play a critical role in the maintenance of neuronal excitability in the postnatal cortex. Acute disruption of ciliary signaling cell-autonomously and rapidly strengthens excitatory synapses onto glutamatergic pyramidal neurons, without affecting inhibitory synapses. Consequently, neurons with disrupted cilia have more and stronger excitatory synapses, and increased mean firing rates. We find that the SSTR3 neuropeptide receptor is selectively localized to the cilia of cortical excitatory neurons, and that inhibition or activation of ciliary SSTR3-mediated signaling bidirectionally modulates excitatory synapses onto these neurons on similar rapid hours-long timescales. Our results indicate that neuropeptidergic signaling via cilia-localized receptors can dynamically modulate neuronal excitability, and raise the possibility that disorders arising from altered E/I balance in cortical circuits may in part be due to defects in cilia function in the postnatal mammalian brain.

We find that acute and cell-autonomous cilia perturbation affects excitatory synapses onto pyramidal neurons, without impacting dendritic inhibitory synapse number, dendritic arborization, intrinsic neuronal excitability, or passive neuronal properties. Thus, this rapid ciliary signaling pathway appears to be specific for the regulation of excitatory synapses, although we are unable to exclude the possibility that the extent of ciliary knockdown may be insufficient to modulate inhibitory synapses on this rapid timescale. There are many neocortical inhibitory interneuron types that synapse onto specific postsynaptic domains of pyramidal neurons (*Kepecs and Fishell, 2014*; *Tremblay et al., 2016*; *Urban-Ciecko and Barth, 2016*), and some classes of inhibitory synapses may be affected. Nevertheless, the net impact of the observed synaptic changes leads to increased firing rates, suggesting that any effects on inhibition also contribute to this enhanced excitability, or are not sufficient to counteract changes in excitatory synaptic drive. It is also currently unknown whether ciliary signaling in neocortical GABAergic interneurons similarly rapidly modulates their synaptic or intrinsic properties. We speculate that cell type-specific ciliary signaling pathways that

integrate distinct extracellular cues may also have important roles in neocortical interneuron subtypes. In this work, we describe the direct and cell-autonomous impact of ciliary signaling in pyramidal neurons. It is likely that the circuit-level impact on excitatory and inhibitory synaptic loops, and thus on circuit excitability, will be more complex in ciliopathies that affect cilia function in all neocortical cell types.

SSTR3 is selectively enriched in the cilia of excitatory neurons, with rare expression in inhibitory neuron cilia. SOM+ interneurons are present throughout cortical layers (*Urban-Ciecko and Barth, 2016*), suggesting that these interneurons may modulate excitatory neurons via cilia localized SSTR3. Since AC3 is present in the cilia of both excitatory and inhibitory neurons, inhibitory neuron cilia are likely to contain a distinct complement of signaling proteins. Neuronal cilia in the brain express Sonic hedgehog (Shh) signaling components, dopamine and serotonin receptor subtypes, as well as subsets of neuropeptide and neurohormone receptors in a brain region- and cell type-specific manner (*Berbari et al., 2008b*; *Brailov et al., 2000*; *Domire et al., 2011*; *Green et al., 2012*; *Hamon, 1999*; *Händel et al., 1999*; *Koemeter-Cox et al., 2014*; *Loktev and Jackson, 2013*; *Sipos et al., 2018*). Adding to this complexity, ciliary localization of these molecules can be dynamically modulated by extracellular signals (*Bangs and Anderson, 2017*; *Domire et al., 2011*; *Green et al., 2016*; *Nachury, 2018*; *Najafi and Calvert, 2012*). Thus, as in olfactory neurons and photoreceptors, central neurons likely actively regulate the targeting and localization of ciliary signal transduction proteins as a function of cell type, developmental stage, and context. This dynamic control of cilia signaling protein content in turn likely allows these organelles to appropriately sense changing extracellular cues and transduce these cues to regulate diverse aspects of neuronal development and function.

There is now increasing evidence of a link between cilia and the establishment and long-term maintenance of synapses in the brain. Prolonged loss of cilia signaling (weeks to months) in multiple brain regions has been shown to result in defects in the maintenance of dendritic and axonal morphology, loss of both excitatory and inhibitory synaptic connectivity, and neuronal degeneration (*Bowie and Goetz, 2020*; *Guo et al., 2017*; *Kumamoto et al., 2012*). In contrast, we find that loss of ciliary signaling on acute timescales of hours to days in cultured neocortical pyramidal neurons rapidly alters excitatory synaptic properties without any apparent effects on dendritic morphology. Our results suggest that neuropeptidergic signaling via ciliary receptors continuously modulates neuronal excitability in the mature brain. In future, precise temporal manipulation of cilia function in vivo may identify the molecular pathways by which ciliary signaling regulates synapse function and maintenance on different timescales.

How might neuropeptidergic signaling from a cilium located on the soma be transmitted to regulate different aspects of synaptic function? Ciliary GPCRs couple with multiple cilia-localized effectors to alter levels of second messengers such as cAMP, calcium, and PI(3,4,5)P3 (*Bielas et al., 2009*; *Delling et al., 2013*; *Garcia-Gonzalo et al., 2015*; *Green and Mykytyn, 2014*; *Guo et al., 2019*; *Hansen et al., 2020*; *Hilgendorf et al., 2016*; *Humbert et al., 2012*; *Moore et al., 2016*; *Mukherjee et al., 2016*; *Mukhopadhyay et al., 2013*; *Mykytyn and Askwith, 2017*; *Schou et al., 2015*; *Su et al., 2013*). These ciliary signals can propagate throughout the cell by as yet unknown mechanisms and activate diverse downstream molecules including the AKT and PKA kinases, as well as transcription factors such as CREB (*Anvarian et al., 2019*; *Bielas et al., 2009*; *Guo et al., 2019*; *Manning and Toker, 2017*; *Mick et al., 2015*; *Moore et al., 2016*; *Mukhopadhyay et al., 2013*; *Plotnikova et al., 2015*; *Tuson et al., 2011*). A role for calcium-, cAMP-, and CREB-mediated changes in gene expression that contribute to some forms of activity-dependent synaptic plasticity is well established (*Flavell and Greenberg, 2008*; *Heinz and Bloodgood, 2020*; *West and Greenberg, 2011*; *Yap and Greenberg, 2018*). Similar transcription-dependent mechanisms may also underlie cilia-driven regulation of network connectivity. On more rapid timescales, ciliary signaling may modulate synaptic strength via posttranslational regulation of synaptic protein function. For instance, cAMP- and calcium-dependent kinases such as PKA and CaMKII modulate synaptic plasticity via direct phosphorylation of AMPAR subunits (*Buonarati et al., 2019*; *Herring and Nicoll, 2016*); the functions of one or both of these kinases at synapses could be modulated by signals from cilia. An important goal for the future will be to identify the ciliary mechanisms and pathways that operate on distinct timescales to modulate synapse establishment, maintenance, and plasticity.

Altered E/I balance in central circuits is linked to a wide range of neurodevelopmental disorders and neuropsychiatric diseases (*Hoftman et al., 2017*; *Nelson and Valakh, 2015*; *Sohal and*

*Rubenstein, 2019*). Intriguingly, many ciliopathies are also characterized by neurological deficits, and altered cilia function and ciliary signaling are associated with defects in neuronal plasticity and circuit functions (*Bennouna-Greene et al., 2011*; *Berbari et al., 2014*; *International Joubert Syndrome Related Disorders Study Group et al., 2008*; *Chen et al., 2016*; *Einstein et al., 2010*; *Guemez-Gamboa et al., 2014*; *Marley and von Zastrow, 2012*; *Rhee et al., 2016*; *Wang et al., 2011*; *Yao et al., 2016*). Moreover, association and linkage studies have identified ciliary genes associated with schizophrenia, autism spectrum disorder, major depressive disorder, bipolar disorder, and others (*Chubb et al., 2008*; *Karunakaran et al., 2020*; *Molecular Genetics of Schizophrenia Collaboration et al., 2008*; *C Yuen et al., 2017*; *Torri et al., 2010*; *Wray et al., 2012*). Cilia are present not only on both excitatory and inhibitory neurons but also on astrocytes which also regulate circuit excitability and synaptic plasticity (*De Pittà et al., 2016*; *Hussaini and Jang, 2018*; *Perez-Catalan et al., 2021*). Our findings raise the possibility that defects in continuous maintenance of E/I balance by ciliary signaling from multiple cell types may underlie a subset of behavioral and cognitive dysfunction linked with mental disorders. Observations reported here establish the cilium as a major modulator of circuit homeostasis in postnatal neurons and highlight the critical importance of future studies of the signaling mechanisms by which this organelle acts in different cells in the brain to regulate neuronal and circuit functions in development, plasticity, and disease.

# Materials and methods

**Key resources table**

| Reagent type (species) or resource | Designation | Source or reference | Identifiers | Additional information |
|---|---|---|---|---|
| Strain, strain background (*Rattus norvegicus*) | Long–Evans | Charles River Laboratories | Strain:006, RRID:RGD_2308852 | |
| Transfected construct (*Rattus norvegicus*) | pAAV-hSyn-EGFP | Bryan Roth via Addgene | RRID:Addgene_50465 | |
| Antibody | αAC3 (Mouse monoclonal) | Encor | Cat#:MCA-1A12, RRID:AB_2744501 | IF(1:500) |
| Antibody | αAC3 (Rabbit polyclonal) | Encor | Cat#:RPCA-ACIII, RRID:AB_2572219 | IF(1:500) |
| Antibody | αARL13B (Mouse monoclonal) | NeuroMab | Cat#:N295B/66 (75-287), RRID:AB_234154 | IF(1:1000) |
| Antibody | αChAT (Goat polyclonal) | Millipore | Cat#:AB144P, RRID:AB_2079751 | IF(1:100) |
| Antibody | αGAD67 (Goat polyclonal) | R and D Systems | Cat#:AF2086, RRID:AB_2107724 | IF(1:2000) |
| Antibody | αGAD67 (Mouse monoclonal) | Millipore | Cat#:MAB5406, RRID:AB_2278725 | IF(1:2000) |
| Antibody | αGluA2 (Mouse monoclonal) | Gift from Gouaux lab, OHSU | | IF(1:1000) |
| Antibody | αIFT88 (Rabbit polyclonal) | ProteinTech | Cat#:13967–1-AP, RRID:AB_2121979 | IF(1:500) |
| Antibody | αPV (Mouse monoclonal) | Synaptic Systems | Cat#:195 011, RRID:AB_2619884 | IF(1:500) |
| Antibody | αShank3 (Guineapig polyclonal) | Synaptic Systems | Cat#:162 304, RRID:AB_2619863 | IF(1:1000) |
| Antibody | αSOM (Mouse monoclonal) | Thermo Fisher | Cat#:14-9751-82, RRID:AB_2572982 | IF(1:500) |
| Antibody | αSSTR3 (Rabbit polyclonal) | Biotrend-USA | Cat#:SS-830–50, RRID:AB_2196357 | IF(1:2000) |
| Antibody | αVGlut1 (Chicken polyclonal) | Synaptic Systems | Cat#:135 316, RRID:AB_2619822 | IF(1:1000) |

*Continued on next page*

*Continued*

| Reagent type (species) or resource | Designation | Source or reference | Identifiers | Additional information |
|---|---|---|---|---|
| Antibody | αVGlut1 (Guineapig polyclonal) | Synaptic Systems | Cat#:135 304, RRID:AB_887878 | IF(1:1000) |
| Antibody | αVIP (Guineapig polyclonal) | Synaptic Systems | Cat#:443 005, RRID:AB_2832228 | IF(1:500) |
| Recombinant DNA reagent | pLKO.1 (plasmid) | David Root via Addgene | RRID:Addgene_10878 | |
| Recombinant DNA reagent | pAAV-hSyn-EGFP (plasmid) | Bryan Roth via Addgene | RRID:Addgene_50465 | |
| Recombinant DNA reagent | pSUPER (plasmid) | OligoEngine | Cat#:VEC-PBS-0002 | |
| Recombinant DNA reagent | pSUPER-H1-shCep164 | This paper | pLRT18 | shRNA: 5'-CAACAACCACATCGAACTTA-3' |
| Recombinant DNA reagent | pLKO-U6-shArl13b_1 | This paper | pLRT19 | shRNA: 5'-CCTGTCAGAAAGGTGACACTT-3' |
| Recombinant DNA reagent | pSUPER-H1-shIft88 | This paper | pLRT26 | shRNA: 5'-CGAATGGCTTGGAGCTTATTA-3' |
| Recombinant DNA reagent | pAAV-H1-shArl13b_2-hSyn-EGFP | This paper | pLRT67 | shRNA: 5'-GCTCAGGACATGATCTCATAA-3' |
| Commercial assay or kit | Zombie Green Fixable Viability Kit | BioLegend | Cat#:423111 | Cell viability assessment |
| Chemical compound, drug | L-796,778 | Gift from Merck Pharmaceuticals | | SSTR3 selective agonist |
| Chemical compound, drug | MK-4256 | MedChemExpress | Cat#:HY-13466 | SSTR3 selective antagonist |
| Chemical compound, drug | Propidium iodide | Thermo Fisher | Cat#:P3566 | |
| Software, algorithm | IGOR Pro | Wavemetrics | RRID:SCR_000325 | https://www.wavemetrics.com/products/igorpro/igorpro.htm |
| Software, algorithm | MATLAB | MathWorks | RRID:SCR_001622 | https://www.mathworks.com/products/matlab.html |
| Software, algorithm | Metamorph | Molecular Devices | RRID:SCR_002368 | http://www.moleculardevices.com/Products/Software/Meta-Imaging-Series/MetaMorph.html |
| Software, algorithm | R (version 4.0.3) | R | RRID:SCR_001905 | https://www.R-project.org/ |
| Software, algorithm | RStudio | RStudio | RRID:SCR_000432 | http://www.rstudio.com/ |

All experimental procedures were approved by the Brandeis IACUC and were performed according to NIH guidelines. All data files used to generate each figure are included as Source Data Files.

## Dissociated cortical neuron cultures and transfection

Dissociated cortical neuron cultures were prepared from visual cortices of male or female P0-3 Long-Evans rat pups and plated on confluent astrocytes as described previously (*Pratt et al., 2003*). Sparse transfections of plasmid DNA were performed after DIV9-10 with Lipofectamine 2000 (Thermo Fisher). GFP-expressing pyramidal neurons were identified by their characteristic morphologies and used for imaging or recording after either 24 hr or 48 hr. Dissociation-matched sister cultures were transfected with vectors expressing GFP alone as controls. All experiments were replicated a minimum of three times from independent dissociations. Data acquisition and analyses were performed blind to treatment conditions.

## Immunofluorescent staining

Cells from DIV11 cultures were fixed with 4% PFA/5% sucrose for 5 min or 15 min and permeabilized with either ice cold methanol for 10 min, or 0.2% Triton X-100 for 5 min. Primary antibodies were applied for either 1 hr at room temperature or overnight at 4°C. To detect GluA2 at the cell

membrane, immunostaining was performed prior to permeabilizing. Primary antibodies included: αARL13B [1:1000, NeuroMab N295B/66 (75-287)], αIFT88 (1:500, ProteinTech 13967–1-AP), αAC3 (1:500, EnCor RPCA-ACIII, MCA-1A12), αSSTR3 (1:2000, Biotrend-USA SS-830–50), αGluA2 (1:1000, gift from Gouaux lab, OHSU), αVGlut1 (1:1000, Synaptic Systems 135 304, 135 316), αShank3 (1:1000, Synaptic Systems 162 304), αPV (1:500, Synaptic Systems 195 011), αSOM (1:500, Thermo Fisher 14-9751-82), αChAT (1:100, Millipore AB144P), αVIP (1:500, Synaptic Systems 443 005), and αGAD67 (1:2000, Millipore MAB5406; 1:2000, R and D Systems AF2086). Secondary antibodies (Thermo Fisher) were incubated for 1–4 hr at room temperature. Slides were mounted using Fluoro-mount-G.

Vectors and shRNAs shRNA sequences were designed with the TRC algorithm (Broad Institute). Vectors used are listed in Key Resources. The pLKO.1 TRC cloning vector was a gift from David Root (RRID: Addgene_10878) (*Moffat et al., 2006*), the AAV-shRNA-ctrl was a gift from Hongjun Song (RRID: Addgene_85741) (*Yu et al., 2015*), and the pAAV-hSyn-EGFP and viral prep 50465-AAV9 were gifts from Bryan Roth (RRID: Addgene_50465). Annealed oligos for shCep164 (5'-CAACAAC-CACATCGAACTTA-3'), shIft88 (5'-CGAATGGCTTGGAGCTTATTA-3'), and shArl13b_2 (5'-GCTCAG-GACATGATCTCATAA-3') were cloned into modified pAAV-shRNA-ctrl or pSUPER vectors (Oligoengine). The sequence of shArl13b_1 was modified from a previously validated shArl13b sequence (5'-CCTGTCAGAAAGGTGACACTT-3') (*Larkins et al., 2011*), and cloned into pLKO.1 and modified pAAV-shRNA-ctrl vectors using Gibson cloning.

## Microscopy and image analysis

Immunostained cells and brain sections were mounted on slides and imaged on either a Zeiss LSM 880 confocal or Zeiss LSM 880 with Airyscan confocal microscope using Plan-Apochromat 63×/1.40 oil objectives. Cilia were labeled with two markers and fluorescence intensity was quantified from ROIs using either Metamorph (Molecular Devices) or ImageJ (NIH). Background fluorescence was subtracted using ROIs from the cell soma or from regions without neurons. Total fluorescence per ROI was averaged and normalized to control treatments. Quantification of synaptic protein intensity and synapse density was performed similar to our published procedures (*Gainey et al., 2015*; *Tatavarty et al., 2020*). Images were taken distal to the primary branch point of apical-like dendrites to ensure uniformity across samples. Analyses of synaptic protein intensity were performed using the Granularity application module in Metamorph; granules with a minimum overlap of 3 pixels in all channels were defined as colocalized puncta and selected for analysis. Total pixel intensities of each punctum were summed and then averaged across puncta for each neuron. For quantification of dendritic complexity, tiled images of apical-like arbors were taken and lengths were measured using ImageJ (NIH); nodes were counted at primary, secondary, and tertiary branch points.

## Electrophysiology

Whole cell patch clamp experiments were performed using an Axopatch 200B amplifier (Molecular Devices) on an Olympus IX70 inverted microscope equipped with differential interference contrast optics and epifluorescence. Recordings were performed at room temperature with an internal solution containing: 120 mM $KMeSO_4$, 10 mM KCl, 2 mM $MgSO_4$, 0.5 mM EGTA, 10 mM HEPES, 3 mM $K_2ATP$, 0.3 mM NaGTP, 10 mM $Na_2$ phosphocreatine; dextrose was used to adjust osmolarity to 320–330 mOsm. Cultures were superfused with artificial cerebral spinal fluid (aCSF) containing: 1 mM $NaH_2PO_4$, 25 mM $NaHCO_3$, 126 mM NaCl, 5.5 mM KCl, 2 mM $MgSO_4$, 2 mM $CaCl_2$; dextrose was used to adjust osmolarity to 330–340 mOsm. GFP-expressing pyramidal neurons were identified by their characteristic morphologies. Neurons with $V_m > -50$ mV, $R_s > 20$ MΩ, $R_{in} < 100$ MΩ, or with $V_m$ or $R_{in}$ changed by ≥10% during the recording were excluded from analysis.

### Spontaneous firing rates

Spontaneous firing rates were recorded in whole cell current clamp mode. A small DC current was injected to keep the resting potential near −55 mV. Ten sweeps of 20 s each were obtained for each neuron and average firing rate was calculated for the entire period of the recording. Spikes were detected automatically using a threshold crossing function written in R (https://github.com/later-eshko/current_clamp_scripts; *Tereshko, 2021a*; copy archived at swh:1:rev: 71bf63383de2b658ae870dba47898b3b784cce79).

## Instantaneous firing rates

F-I recordings were made in whole cell current clamp using depolarizing current steps between 10 and 400 pA in aCSF containing 25 µM picrotoxin, 50 µM APV, and 25 µM DNQX to block synaptic currents. Recordings were acquired with Igor Pro (WaveMetrics) and analyzed as described using custom scripts in MATLAB (MathWorks) (*Joseph and Turrigiano, 2017*). Instantaneous firing rate was calculated as the reciprocal of the interval of the first two consecutive spikes (https://github.com/latereshko/current_clamp_scripts).

## mEPSC recordings

Whole cell voltage clamp recordings were obtained from neurons held at −70 mV. AMPAR-mediated currents were isolated by adding 25 µM picrotoxin, 25 µM APV, and 0.1 µM TTX to aCSF. Events that were <5 pA in amplitude or <3 ms in rise time were excluded from analysis. Recordings were analyzed as described using custom scripts in IGOR Pro (*Joseph and Turrigiano, 2017*; *Tatavarty et al., 2020*) (https://github.com/latereshko/mEPSC_scripts; *Tereshko, 2021b*; copy archived at swh:1:rev:4f77b23d5f94e6e8c5e1a76a059cae4f1811659a).

## SSTR3 pharmacology

The antagonist MK-4256 was purchased from MedChemExpress (HY-13466). The agonist L-796,778 was a generous gift from Merck Pharmaceuticals. For a subset of assays, the agonist was purified via HPLC to isolate the active compound from degraded material (Isaac Krauss, Brandeis University). Compounds were dissolved in DMSO to make 1 mM stock concentrations. DIV10-11 cultures were treated with 0.125 µM, 0.5 µM, 1 µM, or 2 µM concentrations of either reagent for the indicated time periods of 6 hr, 18 hr, or 24 hr. Cells were fixed, immunostained, and imaged as described above.

## AAV viral injections

Virus were diluted in bacteriostatic 0.9% saline on the day of injection (on ice). Prior to surgeries, animals were anesthetized with isoflurane (1.0–2.0% concentration in air) delivered by a SomnoSuite anesthesia system with integrated digital vaporizer (Kent Scientific) through a stereotaxic head holder. Primary visual cortex was bilaterally targeted using stereotaxic coordinates for lambda-bregma distances according to age (P15–16). After craniotomy was performed over the targeted area, a glass micropipette was lowered into the brain and delivered 800 nl of virus-containing solution at the targeted depth. Animals were monitored in separate cages for 12–24 hr post-injection.

## Transcardial perfusions and slices preparation

After 7 days of virus expression, animals (P22–23) were deeply anesthetized with heavy dosage of ketamine/xylazine/acepromazine (KXA) cocktail (140 mg/kg ketamine; 7 mg/kg xylazine; 1.4 mg/kg acepromazine) and perfused with 5 ml of 1× phosphate-buffered saline (PBS) followed by 10–15 ml of 4% paraformaldehyde (PFA) in PBS at room temperature. The brain was removed and preserved in a solution of 4% PFA overnight. After incubation, brain tissues went through three 10 min washes of 1× PBS. Brain tissue was sectioned preserving visual cortex and mounted onto the vibratome with super glue. 50–75 µm sections were collected in a PBS-containing well. Free floating sections were immunostained as described above.

## Cell viability

Cell viability was assessed by co-staining with propidium iodide (Thermo Fisher P3566) and the amine-reactive fluorescent dye Zombie Green (BioLegend 423111). Zombie Green was reconstituted in 100 µl of DMSO. Cultures of neurons were incubated with 50 µg/ml propidium iodide and 1:1000 diluted Zombie Green 10 min before fixation. Cells were fixed as described above. Cells were imaged by confocal microscopy as described above and scored as dead when positively stained for either or both propidium iodide and Zombie Green.

## Statistical analysis

All experiments were replicated a minimum of three times from biologically independent dissociations performed on different days. Data acquisition and analyses were performed blind to treatment

conditions. R software (version 4.0.3) and R Studio were used for statistical analyses (https://www.R-project.org/ and http://www.rstudio.com/). Plots were generated using the package ggplot2 (v3.3.2) (https://cran.r-project.org/web/packages/ggplot2/). Wilcoxon rank-sum or Kruskal–Wallis tests with Dunn's post hoc test for multiple comparisons were used to compare non-normal distributions (dunn.test v1.3.5) (https://CRAN.R-project.org/package=dunn.test). For synaptic protein intensity analyses, values of experimental samples were normalized to the mean values of the control group for each experiment, and linear mixed models (LMM) were used in place of ANOVA to address the non-independence of measurements taken from the same experimental animal or dissociation, using the lme4 package (v1.1–25) (https://cran.r-project.org/web/packages/lme4/). Comparisons were made using random intercept terms for experimental replicate and culture dish, to address variability between preparations and the non-independence of cells imaged from the same dishes. p-values were approximated using the Kenward–Roger method and adjusted with Dunnett's post hoc correction as implemented by the emmeans package (v1.5.2–1) (https://CRAN.R-project.org/package= emmeans). Statistical tests used, p-values, and sample and replicate numbers for each figure are summarized in *Supplementary file 1*. All data and analysis codes can be found at (https://github.com/latereshko/Tereshko_neuron_cilia; *Tereshko, 2021c*; copy archived at swh:1:rev:a975cce55d21d925d6a60157710638e2c54372f4).

## Acknowledgements

We thank Vedakumar Tatavarty for experimental assistance, Kim McDermott and Greg Pazour for advice regarding knockdown of ciliary proteins, and Mike O'Donnell for advice and assistance with R and statistical analyses. We are grateful to Merck for the gift of L-796,778 and to Isaac Krauss for purification of this chemical. We thank the Sengupta and Turrigiano labs for input and advice. This work was funded by the NIH grants R35 GM122463 (PS), R21 MH118464 (PS and GGT), and R35NS111562 (GGT).

## Additional information

### Competing interests

Piali Sengupta: Senior editor, *eLife*. The other authors declare that no competing interests exist.

### Funding

| Funder | Grant reference number | Author |
|---|---|---|
| National Institute of General Medical Sciences | R35 GM122463 | Piali Sengupta |
| National Institute of Mental Health | R21 MH118464 | Gina G Turrigiano Piali Sengupta |
| National Institute of Neurological Disorders and Stroke | R35 NS111562 | Gina G Turrigiano |

The funders had no role in study design, data collection and interpretation, or the decision to submit the work for publication.

### Author contributions

Lauren Tereshko, Data curation, Formal analysis, Validation, Investigation, Visualization, Methodology, Writing - original draft, Writing - review and editing; Ya Gao, Brian A Cary, Investigation; Gina G Turrigiano, Conceptualization, Supervision, Funding acquisition, Visualization, Project administration, Writing - review and editing; Piali Sengupta, Conceptualization, Supervision, Funding acquisition, Writing - original draft, Project administration, Writing - review and editing

### Author ORCIDs

Lauren Tereshko https://orcid.org/0000-0002-3136-753X
Ya Gao http://orcid.org/0000-0002-9608-8988

Brian A Cary (iD) http://orcid.org/0000-0002-1759-164X
Gina G Turrigiano (iD) https://orcid.org/0000-0002-4476-4059
Piali Sengupta (iD) https://orcid.org/0000-0001-7468-0035

## Ethics

Animal experimentation: All experimental procedures were approved by the Brandeis University IACUC (IACUC protocol # 18002) and were performed according to NIH guidelines.

## Decision letter and Author response

Decision letter https://doi.org/10.7554/eLife.65427.sa1
Author response https://doi.org/10.7554/eLife.65427.sa2

## Additional files

### Supplementary files

- Supplementary file 1. Statistical analyses used.

- Transparent reporting form

### Data availability

All data generated or analyzed during this study are included in the manuscript and supporting files. Source data files have been provided for all relevant figures.

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
