## [Decision Letter]

**Acceptance summary:**

This study examines the role of primary cilia in neuronal connectivity and function. Using cultured postnatal rat cortical neurons, the authors elegantly demonstrate that acute disruption of ciliary signaling rapidly strengthens glutamatergic inputs onto cortical pyramidal neurons and increases their spontaneous firing. Bidirectional pharmacological manipulation studies show that ciliary SSTR3 receptor signaling is critical for this effect. This study identifies a role for primary cilia in neuronal function and define a mechanisms that may underlie circuit malfunction in ciliopathies.

**Decision letter after peer review:**

Thank you for submitting your article "Ciliary neuropeptidergic signaling dynamically regulates excitatory synapses in postnatal neocortical pyramidal neurons" for consideration by *eLife*. Your article has been reviewed by three peer reviewers, and the evaluation has been overseen by a Reviewing Editor and Suzanne Pfeffer as the Senior Editor. The reviewers have opted to remain anonymous.

The reviewers have discussed the reviews with one another and the Reviewing Editor has drafted this decision to help you prepare a revised submission.

We would like to draw your attention to changes in our policy on revisions we have made in response to COVID-19 (https://elifesciences.org/articles/57162). Specifically, when editors judge that a submitted work as a whole belongs in *eLife* but that some conclusions require a modest amount of additional new data, as they do with your paper, we are asking that the manuscript be revised if at all possible (as discussed below), and if some aspects cannot be completed in a reasonable amount of time, to either limit claims to those supported by data in hand, or to explicitly state that the relevant conclusions require additional supporting data.

We hope you can include some of the requested additional data with the expectation that if you cannot accomplish all, you will eventually carry out the additional experiments and report on how they affect the relevant conclusions either in a preprint on bioRxiv or medRxiv, or if appropriate, as a Research Advance in *eLife*, either of which would be linked to the original paper.

Summary:

Tereshko et al. broadly examine the role of primary cilia in neuronal connectivity and function. Using cultured postnatal rat cortical neurons, the authors elegantly demonstrate that acute disruption of ciliary signaling rapidly strengthens glutamatergic inputs onto cortical pyramidal neurons and increases their spontaneous firing. Bidirectional pharmacological manipulation studies show that ciliary SSTR3 receptor signaling is critical for this effect. This study identifies a role for primary cilia in neuronal function and define a mechanisms that may underlie circuit malfunction in ciliopathies.

Essential revisions:

The reviewers all found the manuscript to be of broad and substantial interest to cell biologists and neurobiologists and supported its publication in *eLife*. However they also all agreed that two areas of the manuscript would be significantly strengthened with additional data. After discussion about whether additional data were required to support the current claims or whether it was possible to address these points with more discussion, the reviewers agreed that they would state their concerns here and leave it to the authors to decide the manner in which they would address these points. They reiterated in the discussion that they felt the increased impact of the paper with at least some of this additional data would be worth the authors' investment of time.

The first concern was that all of the work relied on neuronal culture. The major concern about this preparation was that because some of the results differed from what has previously been reported in mouse models, it was hard to compare and interpret these findings in light of previous findings. In particular, the fact that the culture preparation has a different synaptic network organization than intact neural circuits was a concern. The reviewers all want to see at least one experiment in a more in vivo circuit context if it were possible for the authors to do in a reasonable time frame. Their comments and suggestions are as follows:

(1) It is unclear to one reviewer how well this recapitulates what happens to these neurons in vivo, especially given that the findings differ from previous findings to some degree (i.e. Guo et al. found that inhibitory synapses are reduced in the striatum; Kumamoto et al. find weakened excitatory synapses; Bowie and Goetz find fewer excitatory connections on Purkinje neurons, etc). There could be very important and interesting reasons for these differences – timescale, cell type, etc as the authors argue, however some attempt should be made to assess the findings in vivo.

(2) Could some of these experiments be done in a context that is closer to in vivo? For example viral transduction of shRNAs into the neonatal cortex followed by an assessment of neuronal connectivity? Electrophysiology might also be performed in slice culture. The authors already use virus to transduce the neonatal cortex in Figure 5, a similar approach could be used to perform in vivo knockdown to confirm some of the results.

(3) Altered E/I balance also occurs when interneuronal cilia function is disrupted (Guo et al., 2017). Highlighting (in Discussion section) how cilia malfunction in interneurons, projection neurons , and associated astroglial cells may converge to contribute to E/I imbalance in cortical circuits in ciliopathies will be useful.

The second area of discussion regarded the role of SSTR3 in ciliary signaling. All of the reviewers felt this was an impactful finding of the study, but all of them also felt that the studies in support of this mechanism could be stronger. Again, the reviewers made suggestions of additional experiments that would strengthen this section and they encourage the authors to consider adding something in this section of the study if it can be done in a reasonable time frame (all three reviewers had one common suggestion – all three versions of the suggestion are included below as 1a-c). The reviewer comments on this section were as follows:

(1a) In Figure 6, the authors conclude that SSTR3 in the primary cilium is primarily important for maintaining appropriate numbers of excitatory synapses. It would be useful to examine whether the SST agonist/antagonist effects are eliminated by the loss of cilia. For example, would the effects of the SST agonist be rescued if cilia were lost?

(1b) Test whether knockdown of the same ciliary proteins affects the response of the neurons to the SST agonist/antagonist. Since SSTR3 is localized almost exclusively to cilia, it seems likely that the effects of the agonist/antagonist are due to signaling at the cilium, and conversely that the increase in excitatory synapses upon cilium ablation is due to loss of SSTR3 from that compartment, but it would be good to take another step to tie those two together more directly.

(1c) In the test of SSTR3, it remains unclear whether the regulation by altering the activity of SSTR3 remains if the cilium is depleted. I feel that both the physiology section (the presentation was immature) and SSTR3 section (the findings were too correlated) are immature. If work hard, these

may be revised within 6-8 weeks.

(2) The authors conclude that “SSTR3 is expressed primarily, if not exclusively, by excitatory neurons in the neocortex” – this statement needs additional evidence. While the images shown in Figure 5 demonstrate the expression of SSTR3 in isolated pyramidal neuronal cilia, unambiguous demonstration of SSTR3 primarily in pyramidal neurons in neocortex will require, at minimum, additional data on SSTR3 ciliary expression in whole cross sections of the neocortex. Images of postnatal neocortex (with all layers) co-labeled with SSTR3 and excitatory or inhibitory neuronal markers will be needed. Including such data will support the conclusion. Otherwise, modify the statement of exclusivity.

---

## [Author Response]

Essential revisions:The reviewers all found the manuscript to be of broad and substantial interest to cell biologists and neurobiologists and supported its publication in eLife. However they also all agreed that two areas of the manuscript would be significantly strengthened with additional data. After discussion about whether additional data were required to support the current claims or whether it was possible to address these points with more discussion, the reviewers agreed that they would state their concerns here and leave it to the authors to decide the manner in which they would address these points. They reiterated in the discussion that they felt the increased impact of the paper with at least some of this additional data would be worth the authors' investment of time.The first concern was that all of the work relied on neuronal culture. The major concern about this preparation was that because some of the results differed from what has previously been reported in mouse models, it was hard to compare and interpret these findings in light of previous findings. In particular, the fact that the culture preparation has a different synaptic network organization than intact neural circuits was a concern. The reviewers all want to see at least one experiment in a more in vivo circuit context if it were possible for the authors to do in a reasonable time frame. Their comments and suggestions are as follows:1) It is unclear to one reviewer how well this recapitulates what happens to these neurons in vivo, especially given that the findings differ from previous findings to some degree (i.e. Guo et al. found that inhibitory synapses are reduced in the striatum; Kumamoto et al. find weakened excitatory synapses; Bowie and Goetz find fewer excitatory connections on Purkinje neurons, etc). There could be very important and interesting reasons for these differences – timescale, cell type, etc as the authors argue, however some attempt should be made to assess the findings in vivo.2) Could some of these experiments be done in a context that is closer to in vivo? For example viral transduction of shRNAs into the neonatal cortex followed by an assessment of neuronal connectivity? Electrophysiology might also be performed in slice culture. The authors already use virus to transduce the neonatal cortex in Figure 5, a similar approach could be used to perform in vivo knockdown to confirm some of the results.

A key difference between observations reported here as compared to previous work (Guo et al., Kumamoto et al., Bowie and Goetz) is the timescale of the manipulations. Here, we report the effects of acute manipulation of ciliary signaling – on the timescale of hours to 1-2 days. In contrast, previous work reported the effects upon disruption of cilia and ciliary signaling on much longer timescales of weeks to months. Further, here we examine neocortical pyramidal neurons while previous studies examined other cell types; there are likely to be cell-type specific roles for ciliary signaling in synapse maintenance. Our results provide the first indication that ciliary neuropeptidergic signaling may continuously modulate excitatory synapses in the cortex. Prolonged loss of this signaling may then lead to disruption of neuronal dendritic and axonal morphology, synaptic connectivity, and neurodegeneration.

Technical challenges prevent us from disrupting ciliary signaling on acute timescales in vivo in the rat cortex. We attempted to knockdown ARL13b in vivo in initial experiments, but found that the earliest time point at which we can detect viral expression is 7 days following virus injection into the rat cortex. Robust viral expression is likely to occur well after 7 days following infection. However, we cannot compare and interpret results obtained from disrupting cilia for >1 week in vivo to those obtained from acutely disrupting cilia in culture for 24-48 hrs. Performing these acute manipulations in the brain remains a major goal for the future but will likely require that we switch to using a mouse model.

We have now added a section to the Discussion that addresses the above points.

3) Altered E/I balance also occurs when interneuronal cilia function is disrupted (Guo et al., 2017). Highlighting (in Discussion section) how cilia malfunction in interneurons, projection neurons , and associated astroglial cells may converge to contribute to E/I imbalance in cortical circuits in ciliopathies will be useful.

We have added text and references in the Discussion that highlight this important point.

The second area of discussion regarded the role of SSTR3 in ciliary signaling. All of the reviewers felt this was an impactful finding of the study, but all of them also felt that the studies in support of this mechanism could be stronger. Again, the reviewers made suggestions of additional experiments that would strengthen this section and they encourage the authors to consider adding something in this section of the study if it can be done in a reasonable time frame (all three reviewers had one common suggestion – all three versions of the suggestion are included below as 1a-c). The reviewer comments on this section were as follows:1a) In Figure 6, the authors conclude that SSTR3 in the primary cilium is primarily important for maintaining appropriate numbers of excitatory synapses. It would be useful to examine whether the SST agonist/antagonist effects are eliminated by the loss of cilia. For example, would the effects of the SST agonist be rescued if cilia were lost?1b) Test whether knockdown of the same ciliary proteins affects the response of the neurons to the SST agonist/antagonist. Since SSTR3 is localized almost exclusively to cilia, it seems likely that the effects of the agonist/antagonist are due to signaling at the cilium, and conversely that the increase in excitatory synapses upon cilium ablation is due to loss of SSTR3 from that compartment, but it would be good to take another step to tie those two together more directly.1c) In the test of SSTR3, it remains unclear whether the regulation by altering the activity of SSTR3 remains if the cilium is depleted. I feel that both the physiology section (the presentation was immature) and SSTR3 section (the findings were too correlated) are immature. If work hard, thesemay be revised within 6-8 weeks.

All reviewers requested that we determine whether loss of cilia abolishes SSTR3-dependent effects on excitatory synapses.

To address these comments, we transfected cultures with shIft88/shCep164 or GFP alone for 48 hrs, treated with the SSTR3 agonist or DMSO (control) for 24 hrs, and then immunostained for Shank3 (postsynaptic marker) and SSTR3. As shown in new Figure 6D, a subset of the truncated cilia in shRNA-transfected neurons retained expression of SSTR3. Consistent with our previous data (Figure 6B), we find that treatment with the agonist decreased Shank3 intensity in control (GFP-expressing) neurons (new Figure 6E). As expected from our data with truncated cilia (Figure 2C), reducing cilia length increases Shank3 intensity (new Figure 6E). Importantly, in neurons with truncated cilia, the SSTR3 agonist was no longer able to reduce Shank3 intensity (new Figure 6E). These results imply that the cilium is necessary for SSTR3 signaling-dependent regulation of excitatory synapses.

2) The authors conclude that “SSTR3 is expressed primarily, if not exclusively, by excitatory neurons in the neocortex” – this statement needs additional evidence. While the images shown in Figure 5 demonstrate the expression of SSTR3 in isolated pyramidal neuronal cilia, unambiguous demonstration of SSTR3 primarily in pyramidal neurons in neocortex will require, at minimum, additional data on SSTR3 ciliary expression in whole cross sections of the neocortex. Images of postnatal neocortex (with all layers) co-labeled with SSTR3 and excitatory or inhibitory neuronal markers will be needed. Including such data will support the conclusion. Otherwise, modify the statement of exclusivity.

We now provide representative images of neurons in all layers of the V1 area of the neocortex of P1 rats co-immunostained for NeuN, GAD67, and SSTR3 in new Figure 5—figure supplement 1A. Under our conditions, NeuN primarily (although not exclusively) marks excitatory neurons, whereas GAD67 exclusively marks inhibitory neurons. As shown in new Figure 5—figure supplement 1A, B, while the majority of NeuN-positive neurons across all layers contain ciliary SSTR3, fewer than 10% of GAD67-positive neurons in each cortical layer express this receptor although the cilia of these inhibitory interneurons continue to express AC3 (Figure 5C, and Figure 5—figure supplement 1C). These data further support our conclusion that ciliary SSTR3 expression is largely restricted to excitatory neurons in the neocortex.

We also added new data to Figure 5D showing that VIP+ interneurons do not contain SSTR3-positive cilia.